# Multi-Sensor-Based Blind-Spot Reduction Technology and a Data-Logging Method Using a Gesture Recognition Algorithm Based on Micro E-Mobility in an IoT Environment

**DOI:** 10.3390/s22031081

**Published:** 2022-01-30

**Authors:** Hyoduck Seo, Hyeonbo Kim, Kyesan Lee, Kyujin Lee

**Affiliations:** 1College of Electronics & Information, Kyunghee University, 1732 Deogyeong-daero, Giheung-gu, Yongin-si 17104, Gyeonggi-do, Korea; kyesan@khu.ac.kr; 2Vehicle Research Team, Gyeongbuk Institute of IT Convergence Industry Technology, 1, Gongdan 9-ro 12-gil, Jillyang-eup, Gyeongsan-si 38463, Gyeongsangbuk-do, Korea; hbkim@gitc.or.kr; 3Department of Electronic Engineering, Semyung University, 65 Semyung-ro, Jecheon-si 27136, Chungcheongbuk-do, Korea

**Keywords:** micro e-mobility vehicle (MEV), gesture recognition algorithm, blind spot, data logging

## Abstract

Autonomous driving is evolving through the convergence of object recognition using multiple sensors in the fourth industrial revolution. In this paper, we propose a system that utilizes data logging to control the functions of micro e-mobility vehicles (MEVs) and to build a database for autonomous driving with a gesture recognition algorithm for use in an IoT environment. The proposed system uses multiple sensors installed in an MEV to log driving data as the vehicle operates and to recognize objects surrounding the MEV to remove blind spots. In addition, the proposed system is capable of multi-sensor control and data logging for the MEV based on a gesture recognition algorithm, and it can provide safety information to allow the system to address blind spots or unexpected situations by recognizing the appearances or gestures of pedestrians around the MEV. The proposed system can be applied and extended in various fields, such as 5G communication, autonomous driving, and AI, which are the core technologies of the fourth industrial revolution.

## 1. Introduction

The arrival of the fourth industrial revolution does not eliminate the core technologies that arose and were developed from the first industrial revolution through the third industrial revolution or its older technology, but this period can be considered an era of new convergence technology through coexistence and fusion with existing core technologies [1]. There are several interesting characteristics of new convergence technologies. First, they are not limited to specific fields but change existing production modes by inducing technological innovations in various fields. Second, various complementary studies and innovations using the new technology paradigm are carried out in a series over a long period of time. Examples of this process include steam engines of the first industrial revolution, electric technology of the second industrial revolution, and computers and internet-based information technology of the third industrial revolution [1].

As shown in Figure 1, it is technology that constitutes and leads the fourth industrial revolution, and various advanced technologies, such as artificial intelligence and robotics, the Internet of Things, autonomous vehicles, 3D printing, nanotechnology, biotechnology, material engineering, energy storage technology, and ubiquitous computing, have arisen [2] during this period. In particular, among these technologies, those in the fields of physics and biology have evolved one step further through mutual exchanges and convergence with digital technologies, while technological changes in other fields have already reached the inflection point of development due to the exponential rate of development [3].

An eco-electric vehicle with autonomous driving, one of the core technologies drawing attention during the fourth industrial revolution, is an example of a new type of convergence technology. 

Eco-friendly electric vehicles are the focus of the paradigm of automobile technology changes emerging in the fourth industrial revolution. As shown in Figure 2, the paradigm shift for vehicles is continuing with the emergence of various technologies, ranging from steam vehicles to eco-friendly autonomous vehicles. With the transition from fossil fuels to electricity, autonomous driving is also developing through convergence with various technologies, such as multi-sensor, vehicle software, and artificial intelligence, which are not applied to existing vehicles [4].

Autonomous driving is defined as driving technology that allows a vehicle to travel to a destination on its own, without the need for a driver to operate it [5]. As such, autonomous driving is evolving from existing automobile technology to a new type of convergence technology through the convergence of automobiles and various sensors (e.g., cameras and LIDAR) that became universalized between the first industrial revolution and the third industrial revolution [6].

However, all types of vehicles have blind spots. In addition, MEVs have a serious problem with blind spots, which are smaller in size and have fewer specification resources than conventional vehicles. Figure 3 shows the blind spot of an MEV. The green area is a visible area, where it is possible to recognize surrounding objects. However, the red area is defined as a blind spot, where objects cannot be recognized. Blind spots can lead to critical traffic accidents.

In addition, while autonomous driving continues to receive much attention, many problems must be solved in order to disseminate and commercialize this technology. Conventional autonomous driving has a disadvantage in that its reliability is determined by external environmental conditions. Furthermore, one existing issue with the use of object recognition technology in autonomous driving based on multiple sensors is that accidents involving fatalities often occur through misrecognition of pedestrians or vehicles that arise when vehicles, pedestrians, and traffic facilities are recognized as identical. These problems hinder the popularization of autonomous driving.

Accordingly, in this paper, in an effort to resolve these issues and improve the reliability of autonomous driving, we propose a method involving gesture recognition combined with multi-sensor data logging. The gesture recognition algorithm used with the proposed system expands the range of gesture recognition to control various recognition data from multiple sensors, unlike existing gesture recognition algorithms that control only infotainment functions in vehicles for the convenience of drivers and passengers. As autonomous vehicles are further developed, the structure of advanced vehicles will become increasingly simplified. Conventional control devices such as steering wheels or gear shifts will disappear, which will advance the method of vehicle control through the motion or behavior of passengers. Therefore, the accuracy of the gesture recognition system of the conventional vehicle should be improved, and it will become the most important function to control the vehicle. Moreover, the proposed system can reduce the amount of misrecognized data by improving the object recognition rate through an extended gesture recognition algorithm for various objects recognized by multiple sensors. The proposed data-logging method establishes a database based on multi-sensor data to improve the accuracy of the gesture recognition algorithm and the safety of autonomous driving. The data constructed with various gestures will be used as reference data to improve the accuracy of the gesture recognition algorithm, and various driving environment data will be used as learning data to improve the safety of autonomous driving. Therefore, in this paper, we also propose a logging system that compiles reference data for autonomous driving by logging data from an extended gesture recognition algorithm and multiple sensors in an IoT environment to improve the reliability of autonomous driving.

## 2. Proposed System

The proposed system uses multiple sensors installed in the MEV to log driving data as the device operates. The system recognizes objects surrounding the MEV to remove blind spots in an IoT environment.

Figure 4 shows a conceptual diagram of the proposed system. The proposed system can control multiple sensors installed in the MEV based on a gesture recognition algorithm while acquiring driving data as the MEV is driving. In addition, multiple sensors installed in the MEV recognize surrounding objects and notify the MEV system. By recognizing the gestures or appearances of pedestrians, information is provided to the MEV so that it can address the situation. Therefore, the proposed system is capable of multi-sensor control and logging driving data for an MEV based on a gesture recognition algorithm, and it can provide safety information to allow the system to manage blind spots or unexpected situations by recognizing appearances or gestures of pedestrians around the MEV. 

### 2.1. Proposed Gesture Recognition Algorithm

Figure 5 shows the structure of the proposed gesture recognition system. This system consists of a gesture recognition algorithm, an infotainment system for the MEV, and an IoT communication module for controlling multiple sensors (e.g., GPS, LIDAR, and camera) installed in the MEV. The gesture recognition algorithm defines commands with various hand gestures. Each hand gesture is defined as shown in Figure 6 based on Figure 7 to control the infotainment of the MEV. Gesture recognition is recognized in the field of view (FOV) of the 3D depth camera. Skeleton modeling of recognized hand gestures is applied to the recognition algorithm. The gesture recognition algorithm processing process is described in detail in Section 3. The gesture recognition system is applied between the interior mirror and the front seat lighting to recognize the hand gestures of the MEV user and to control the infotainment system and multiple sensors (e.g., GPS, LIDAR, and camera) of the MEV in an IoT environment. The proposed system is based on a 3D depth camera (RealSense D435 by Intel), which acquires 3D data [7]. The gesture recognition algorithm includes a detection step for defining the user’s hand region in the acquired data and a gesture recognition step for recognizing actual behavior by the driver. The recognized gesture image from the 3D depth camera is separated from the background and the hand area, and the user’s hand is modeled using color image information [8]. In the gesture recognition step, the event sequence is designed to determine the user’s input with a random decision forest-based recognizer based on the modeling result of the detected hand [9]. The recognition results from the predefined gestures and from additional information on the vehicle are communicated to the vehicle system in the IoT environment.

Figure 6 presents examples of gestures as defined by the gesture recognition algorithm. The user’s gestures can serve to define other gestures as needed. In addition, by defining the gestures or movements of pedestrians recognized by the multiple sensors, particularly the cameras and LIDAR sensors, safety can be ensured by providing information to the user of the MEV, even in dangerous or unexpected situations caused by blind spots.

Figure 7 shows a scenario in which various gestures defined in Figure 6 serve to control the vehicle. Gestures are defined as static gestures according to the shape of the hand and dynamic gestures according to the spatial movement of the hand, and they are intuitively and dynamically defined to avoid an overly complicated gesture recognition algorithm.

### 2.2. Multi-Sensor-Based Data Logging for Autonomous Driving

The multi-sensor-based data-logging system refers to a type of technology that allows the control of multiple sensors (GPS, LIDAR, and camera) based on the proposed gesture recognition algorithm and acquires driving environment data for the MEV in the IoT environment [10].

Figure 8 shows the MEV with multiple sensors (GPS, LIDAR, and camera) installed to compile driving data. Also, the red circle shown in Figure 8 means each multi-sensor. By installing cameras on both side mirrors and on the rear of the MEV, objects on the side and rear of the MEV are recognized in the form of 2D image data and constructed as driving data. LIDAR sensors are installed on both sides of the rear bumper of the MEV to recognize objects on the sides and rear of the device as 3D image data and are compiled into driving data [11]. In addition, a GPS receiving module is installed inside the MEV, and a GPS receiving antenna is installed on the ceiling to acquire driving routes and speed data of the MEV and compile it into driving data as well.

Figure 9 shows the multi-sensor-based driving logging system using RTMap. The RTMap system can be converted into data such as images and CSV so that various forms of data acquired by the multiple sensors of the MEV, installed as shown in Figure 8, can be used. The proposed logging system constructs driving data by dividing the real-time driving data of the MEV into image data from the camera and LIDAR sensors and location information data from GPS. It obtains data from GPS, which is able to construct CSV files and images using the latitude, longitude, and altitude. In addition, it obtains other types of data, which can be displayed with a text viewer on the proposed logging system screen. The data acquired from the camera are compiled in JPEG format after the detection of objects on the side and rear of the MEV in the form of 2D image data, with real-time images then displayed on the screen image viewer of the proposed logging system. The data acquired from the LIDAR sensors are compiled in CSV format after detecting the side and rear objects surrounding the MEV in the form of 3D image data. These data are also displayed through the 3D viewer of the proposed logging system.

## 3. Results

Figure 10 presents the results of verification of the performance of the gesture recognition algorithm. Figure 10a shows a color image of data acquired from the 3D depth camera, a type of gesture detection sensor. Figure 10b shows the depth image data acquired from the 3D depth camera. Figure 10c,d contain depth image data of an object acquired within the effective recognition distance of the 3D depth camera. Figure 10e,g visualize the area of a gesture in a color image defined in Figure 10c,d. The gesture area is defined in Figure 10g from the gesture area and background area separated in Figure 10f,g. The input gesture shape maps the designed gesture command by the proposed gesture algorithm when obtaining the shape of the gesture in Figure 10g. The recognized objects shown in Figure 10 are the palms (backs of the hands), wrists, and fingers of the hand, and each defined area is visualized for object separation and feature extraction. In Figure 10h, objects created within the effective recognition distance are visualized based on the extracted features for each object. The object used in Figure 10 is a hand, and this result can be obtained as a visualization model of a hand recognized within the effective recognition distance of the gesture recognition algorithm.

Figure 11 shows the results of various gestures defined in the gesture recognition algorithm through the hand model generation process. Various types of gestures were defined, the model was visualized by detecting the defined gestures within the effective recognition distance, and the performance result of the gesture recognition algorithm was confirmed to show a high recognition rate.

Figure 12 shows the mapping result from gesture recognition and the definition command. In the gesture recognition algorithm, commands (e.g., Go to Home Menu, Next Menu, and Prev. Menu) were defined for each gesture to determine the presence or absence of motion, and it was confirmed that the simulation result can be accurately mapped and controlled.

Figure 13 presents the results screen of the logging system, showing the data acquired by the multiple sensors (i.e., GPS, LIDAR, and camera) in the IoT environment. The results of the logging system show data obtained while the MEV was driving in real time. The 3D image view in the red square shows the data from the LIDAR sensors installed in the MEV, and the video view in the blue square shows the data from the camera installed in the device. In Figure 13, only the rear camera is shown, but data from both cameras in the side mirror can also be displayed, and data from all cameras can be displayed in a single view. Additionally, the green square shows the GPS data installed in the MEV with the text viewer and map image viewer. The text viewer displays various aspects of GPS information, such as the latitude, longitude, and altitude.

Figure 14 and Figure 15 show the results after compiling data acquired from the multi-sensor-based logging system in the IoT environment. Figure 14 provides basic information for detecting objects recognized from the side and rear of the MEV based on data acquired from the LIDAR sensors and cameras installed in the MEV and for identifying the types of objects. Objects recognized by LIDAR are used to define the types by means of the camera, and the blind spot information can be provided to the user of the MEV as well. Therefore, the proposed logging system can function to eliminate blind spots by providing object and state information pertaining to blind spots based on data acquired from the camera and LIDAR sensors. Figure 15 shows information related to driving, such as the travel route and mileage of the MEV, based on data acquired from the GPS installed in the MEV. By means of the proposed logging system, it is possible to understand the environment or traffic conditions of the road based on actual driving information acquired from GPS. This can also be provided as driving correction data that can improve the autonomous driving algorithm based on driving information data acquired from GPS and 2D image data acquired from the cameras.

As a result, the performance of the gesture recognition algorithm of the proposed system and the driving data-logging system of the multi-sensor-based MEV used in an IoT environment could be confirmed through a simulation. The proposed system does not implement and verify the two technologies separately; instead, it requires the convergence of the two technologies. Therefore, the proposed gesture recognition algorithm expands the range of gesture recognition so that it can be a means by which to control the multiple sensors (GPS, LIDAR, and camera) installed in the small electric vehicle, not merely the indoor infotainment functions of the vehicle. As a result, it was possible to reduce the amount of data related to misrecognized objects through gesture recognition for data recognized by multiple sensors. In addition, by recognizing a gesture or a specific motion of an external pedestrian, the occupant can prepare for risks or unexpected situations occurring in the blind spots of a miniature electric vehicle. Therefore, the convergence of the gesture recognition algorithm and autonomous driving constituting the proposed system defines various gestures and motions in the IoT environment, enabling control and communication not only in an autonomous driving environment but also in a variety of conditions. Moreover, through the proposed system, it will be possible to popularize autonomous driving technology by reducing instances of misrecognized object data based on the gesture recognition algorithm and thus helping to improve the safety and reliability of autonomous driving.

## 4. Conclusions

In the method proposed in this paper, recognized object data are processed using a proposed gesture recognition algorithm based on multiple sensors. The presented algorithm reduces instances of misrecognition by using multiple sensors, leading to much more accurate recognition. In addition, unexpected situations occurring in the blind spots of an MEV can be recognized by the occupants through the object recognition processes of multiple sensors. Given the recognized data and situation, the danger from unexpected situations occurring in blind spots is mitigated, and safety can be improved for the occupants and the external environment through the proposed gesture recognition algorithm. 

Although the proposed system has to analyze the results in the practical MEV environment, the proposed system analyzed the results in a laboratory environment. Because gesture recognition systems have not been implemented in the practical MEV environment, there are no previous results to compare with the system proposed in this paper. Therefore, in this study, we analyzed the results of the gesture recognition algorithm in a laboratory environment. Furthermore, in future research, we plan to analyze the results of the proposed system in a practical MEV environment, which was not performed in this study.

Moreover, the proposed system complements the shortcomings of existing autonomous driving, which is limited by the road environment, through convergence with autonomous driving, thus providing high reliability through data logging and gesture recognition algorithms that process data recognized in various road environments. An autonomous driving reference database that can improve safety can also be compiled.

In addition, the proposed system is not limited to controlling MEVs and autonomous driving but can also be applied to various other fields. For example, it can be applied to a network environment in which various IoT systems, such as those in homes and offices, exist. It is also possible to control IoT-based smart devices and sensors by defining a gesture recognition range from data recognized in an environment in which IoT-based smart devices and sensors are utilized. As such, the proposed system can be utilized in various fields and in conjunction with a range of technologies. These advantages enable innovations in the form of new algorithms and technologies through convergence with technologies in various fields, such as 5G, autonomous driving, and AI, which are the core technologies of the fourth industrial revolution.

## Figures and Tables

**Figure 1 sensors-22-01081-f001:**
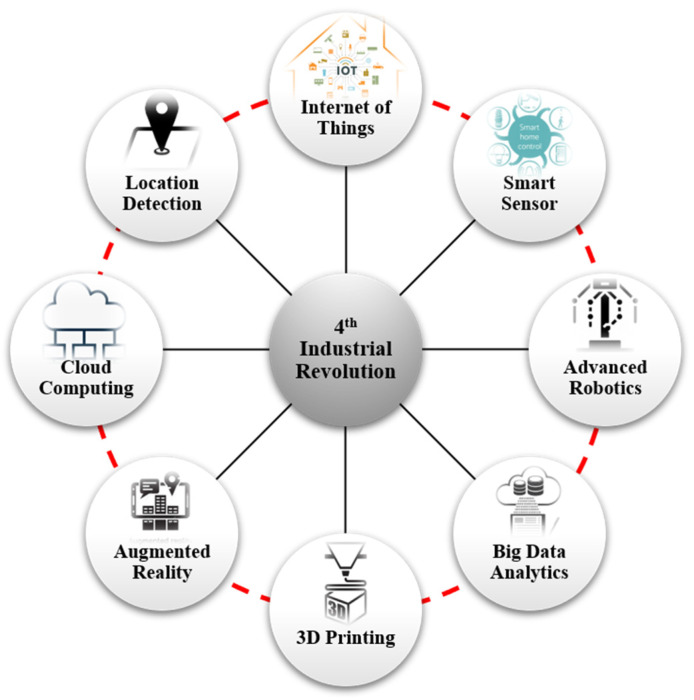
Various advanced technologies that constitute the fourth industrial revolution.

**Figure 2 sensors-22-01081-f002:**
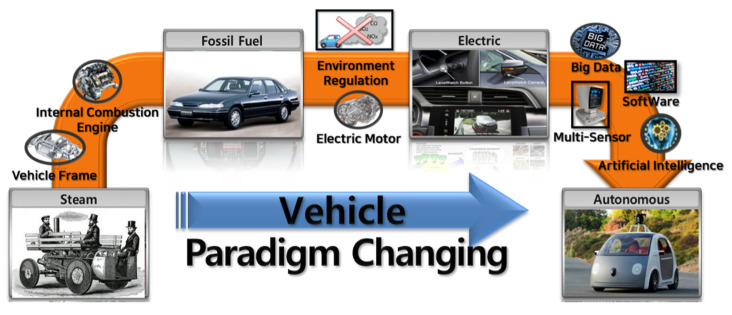
Paradigm change for vehicles (steam to autonomous).

**Figure 3 sensors-22-01081-f003:**
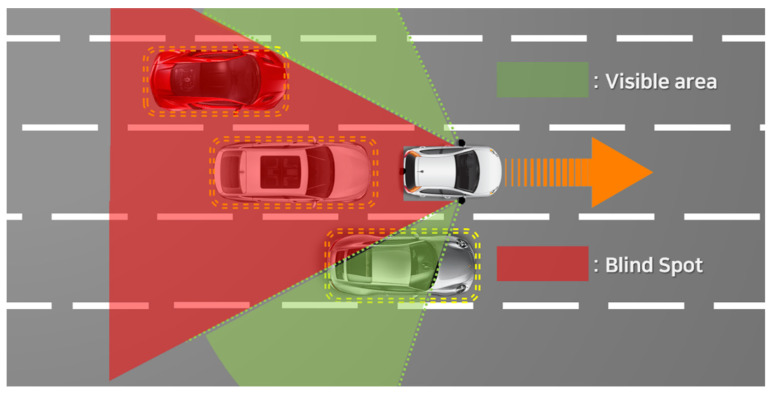
Blind-spot problem in micro e-mobility vehicle (MEV).

**Figure 4 sensors-22-01081-f004:**
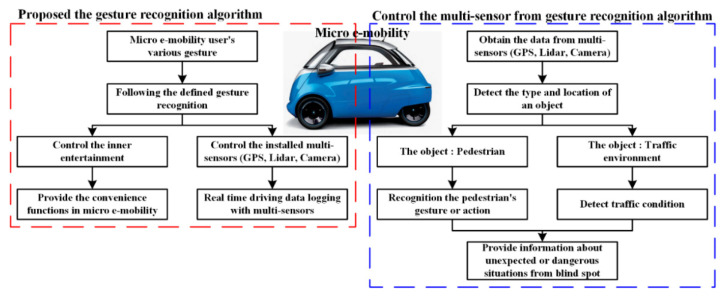
Conceptual diagram of the proposed system.

**Figure 5 sensors-22-01081-f005:**
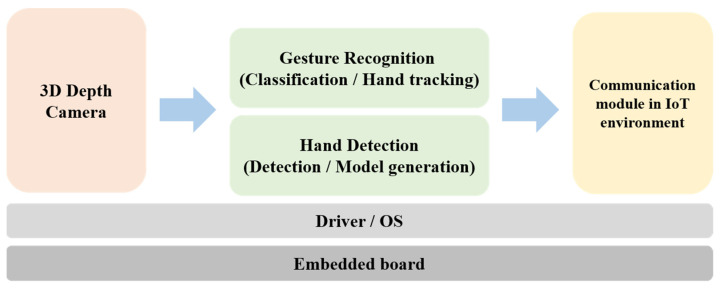
Configuration diagram of the proposed gesture recognition system.

**Figure 6 sensors-22-01081-f006:**
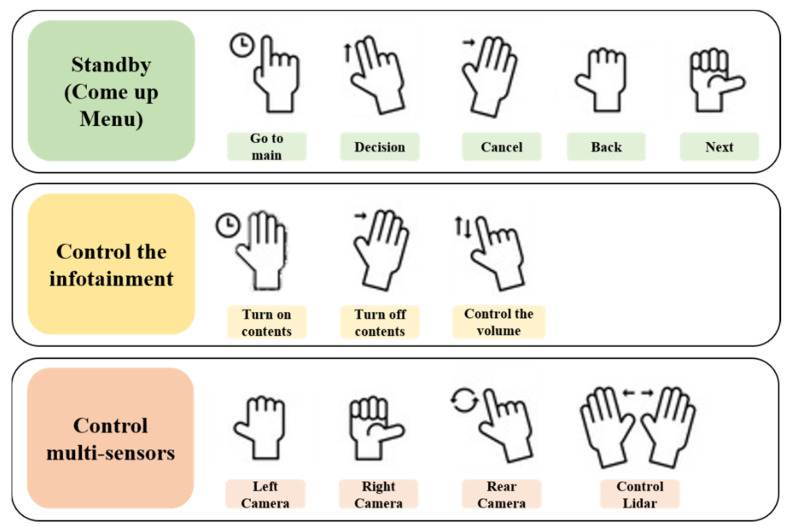
Example scenario based on the recognition of various gestures.

**Figure 7 sensors-22-01081-f007:**
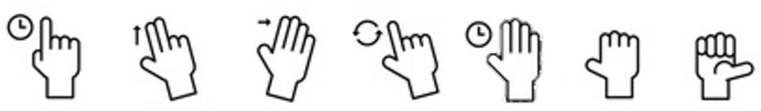
Various gesture definitions for recognition.

**Figure 8 sensors-22-01081-f008:**
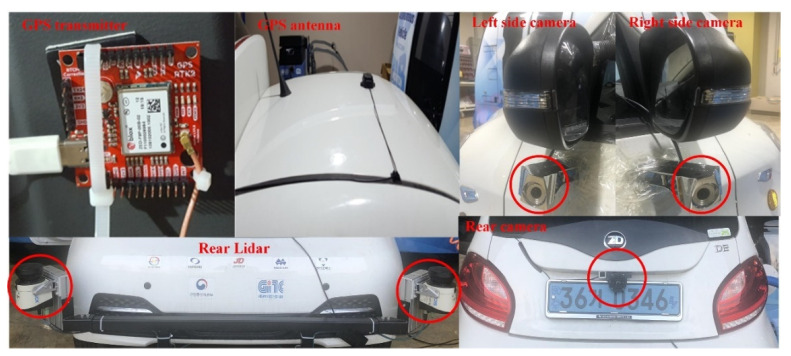
Multiple sensors (GPS, LIDAR, and camera) installed in the micro e-mobility vehicle.

**Figure 9 sensors-22-01081-f009:**
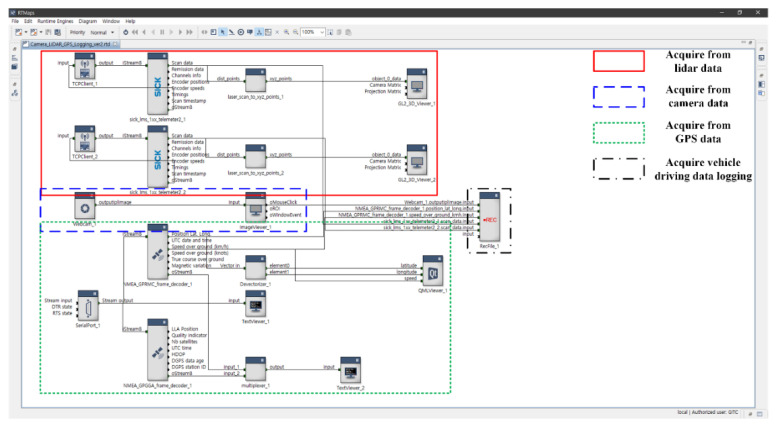
Data-logging system based on multiple sensors (GPS, LIDAR, and camera) using RTMap.

**Figure 10 sensors-22-01081-f010:**
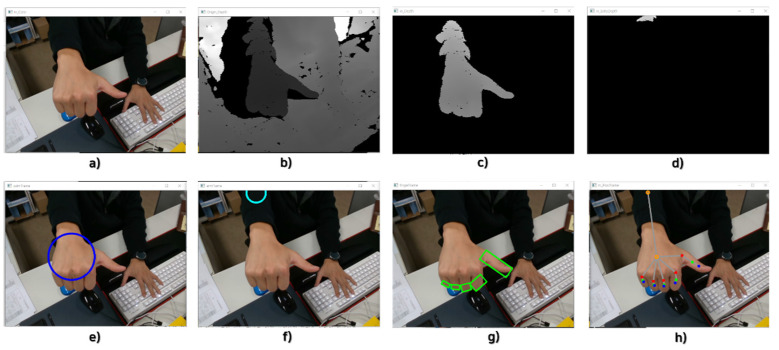
Gesture recognition algorithm detection process and results (Gesture Type 1).

**Figure 11 sensors-22-01081-f011:**
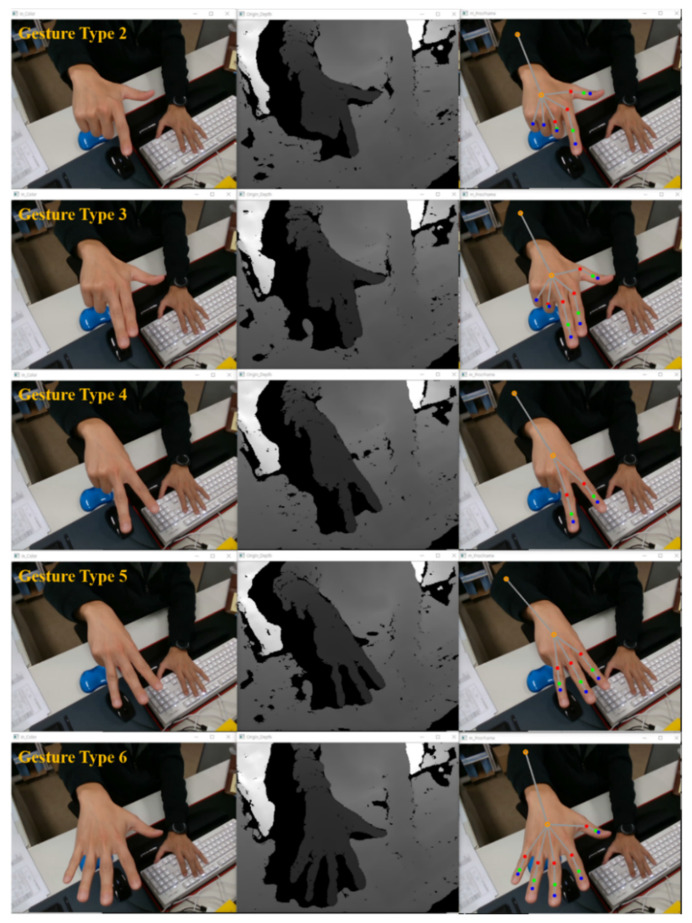
Various gesture detection results using the gesture recognition algorithm (Gesture Types 2~6).

**Figure 12 sensors-22-01081-f012:**
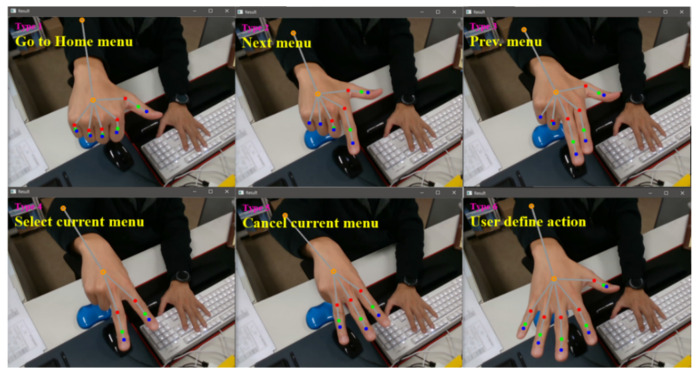
Gesture-type judgment results obtained through the gesture recognition algorithm.

**Figure 13 sensors-22-01081-f013:**
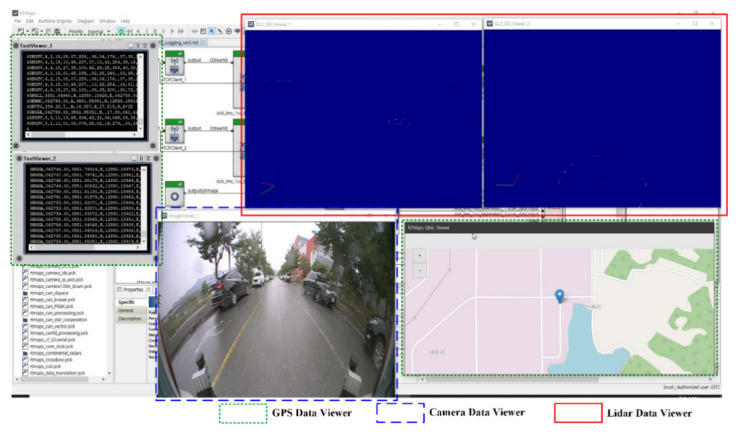
Real-time driving data results using the logging system with multiple sensors (GPS, camera, and LIDAR).

**Figure 14 sensors-22-01081-f014:**
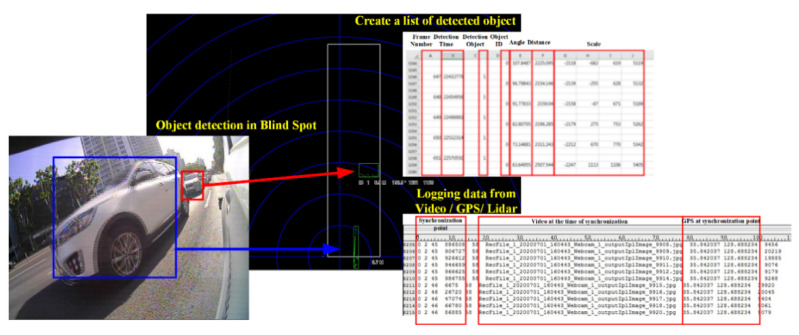
Object detection results using data acquired from GPS, camera, and LIDAR sensors of the proposed system.

**Figure 15 sensors-22-01081-f015:**
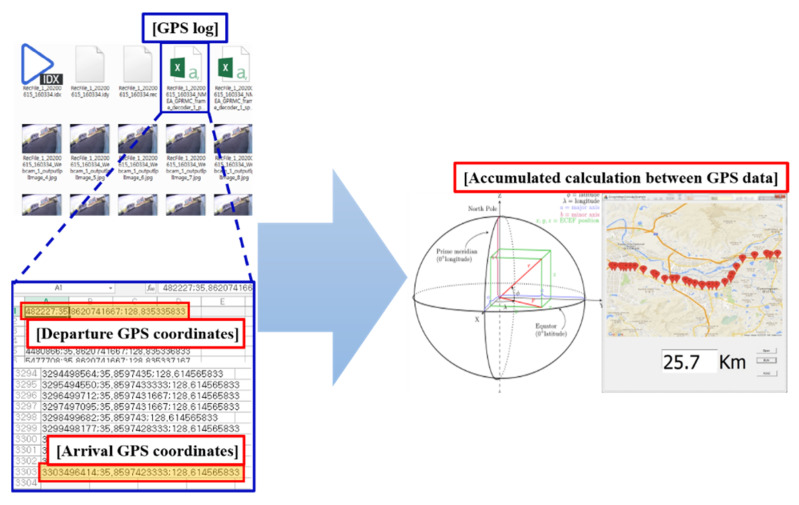
Driving data acquired from GPS in the proposed logging system.

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
