# Peer review of "Multi-Sensor-Based Blind-Spot Reduction Technology and a Data-Logging Method Using a Gesture Recognition Algorithm Based on Micro E-Mobility in an IoT Environment"

_sensors, 2022, doi:10.3390/s22031081_

Round 1

Reviewer 1 Report

The title mentions “Multi-sensor-based blind-spot reduction technology” as if it is a contribution to be found in this paper. But there is no proposed contribution or pertinent section in the paper. The only reference to the blind-spot reduction technology is in the Results section.

There is no related work section and the introduction gives very little reference to related work (only five generic references found).

The performance of the proposed gesture recognition approach is not quantitatively evaluated nor compared with any other approach.

The proposed gesture recognition approach is shown to be tested only with in-office images and not in a realistic scenario in the vehicle.

The term “micro e-mobility vehicle” is repeated many times in the manuscript and at many cases leads to long sentences that make comprehension difficult. Consider using an acronym for this term.

The results in terms of data-logging correctness are not evaluated in terms of correctness of accuracy.

The text in figure 8 is almost unreadable even in magnification.

Figure 4 should denote the flow of information or interaction between the illustrated modules.

The proposed contribution of the paper is not clear in the introduction. The reader has to reach Section 2 to find out. Also the purpose of data logging should be made clear in the introduction too.

The two figures in the introduction and the review on the four industrial revolutions is not necessary. I would rather see the space used to provide an illustration and better description of what this work is about and what the contents of this paper are.

Reviewer 2 Report

The manuscript entitled “Multi-sensor-based blind-spot reduction technology and a data-logging method using a gesture-recognition algorithm based on micro e-mobility in an IoT environment”, introduces a data logging method to control the functions of micro e-mobility vehicles for autonomous driving with a gesture-recognition algorithm in an IoT environment. The novelty of this research is low, and the contribution is confusing. A thorough revision is required before considering this work-

  1. Section 2.1 needs to be improved. The author mentioned a proposed algorithm for gesture recognition; however, there is no detailed explanation. Please explain the contribution more scientifically.
  2. It is mentioned in the abstract that “In this paper, we propose a base technology for…”. What does it mean by “base technology”?
  3. The introduction section is poor. Authors need to focus on the specific topic rather than unnecessary issues like the “industrial revolution”. The author also needs to enhance the motivation and discuss some related work such as (since this work is based on gesture interaction) – Gesture recognition (https://ieeexplore.ieee.org/document/9404361/), air-writing (https://ieeexplore.ieee.org/abstract/document/7322243), etc. How the proposed work contributes to the body of the knowledge. Additionally, the references are not adequate.
  4. The author mentioned in the text-“CAN communication protocol.”, but did not explain about CAN.
  5. Revise the Figure 1 caption and labels.
  6. The representation of Figure 9 is confusing. The author mentioned – “Figures. 9-5, 9-6, and 9-7 visualize the area of a gesture in a color image as defined in Figures. 9-3, and 9-4.” I would suggest writing like – 9(a), 9(b), etc.
  7. The whole paper needs to be revised in terms of readability. Some examples (not limited to) –
    1. “In this paper, we propose a base technology for the diffusion of autonomous driving technology, one of the leading technologies of the fourth industrial revolution, and for the dissemination of various vehicles.”
    2. “Figure 5 shows examples of various gestures defined based on the gesture-recognition algorithm.”
    3. “Data acquired from GPS are constructed as CSV files and images using the latitude, longitude, and altitude, and other types can be displayed with a text viewer on the proposed logging system screen.”

Round 2

Reviewer 1 Report

I find that the authors did a very good job in improving the presentation of their work and as a result, the paper is much more comprehensible.

As the gesture recognition experiments were only preliminary and in-lab evaluated, I would recommend that authors discuss this in their conclusion and point to a future work that would evaluate this system in realistic situations.

Author Response

Thank you for your review.

I think this paper is possible to more improve than previous.

Reviewer 2 Report

The manuscript entitled “Multi-sensor-based blind-spot reduction technology and a data-logging method using a gesture-recognition algorithm based on micro e-mobility in an IoT environment”, introduces a data logging method to control the functions of micro e-mobility vehicles for autonomous driving with a gesture-recognition algorithm in an IoT environment. The novelty of this research is low, and the contribution is confusing. 

The revised version sounds good. However, the main concern was the methodological part. I think significant improvement is still necessary.

Author Response

(The authors gave the same response as above.)
